# Resistance to Pyrethroids in the Malaria Vector *Anopheles albimanus* in Two Important Villages in the Soconusco Region of Chiapas, Mexico, 2022

**DOI:** 10.3390/ijerph20054258

**Published:** 2023-02-27

**Authors:** Francisco Solis-Santoyo, Cuauhtémoc Villarreal-Treviño, Alma D. López-Solis, Lilia González-Cerón, José Cruz Rodríguez-Ramos, Farah Z. Vera-Maloof, Rogelio Danis-Lozano, Rosa Patricia Penilla-Navarro

**Affiliations:** 1Centro Regional de Investigación en Salud Pública, Instituto Nacional de Salud Pública, Cuarta Norte y 19 Calle Poniente, Centro S/N, Tapachula CP 30700, Chiapas, Mexico; 2Jurisdicción Sanitaria VII, Antiguo Hospital General de Tapachula, Carretera Antiguo Aeropuerto, Tapachula CP 30798, Chiapas, Mexico

**Keywords:** *Anopheles albimanus*, insecticide resistance, pyethroids, imported malaria, esterases, forced oviposition

## Abstract

Chiapas State comprises the largest malaria foci from Mexico, and 57% of the autochthonous cases in 2021, all with *Plasmodium vivax* infections, were reported in this State. Southern Chiapas is at constant risk of cases imported due to migratory human flow. Since chemical control of vector mosquitoes is the main entomological action implemented for the prevention and control of vector-borne diseases, this work aimed to investigate the susceptibility of *Anopheles albimanus* to insecticides. To this end, mosquitoes were collected in cattle in two villages in southern Chiapas in July–August 2022. Two methods were used to evaluate the susceptibility: the WHO tube bioassay and the CDC bottle bioassay. For the latter, diagnostic concentrations were calculated. The enzymatic resistance mechanisms were also analyzed. CDC diagnostic concentrations were obtained; 0.7 μg/mL deltamethrin, 12 μg/mL permethrin, 14.4 μg/mL malathion, and 2 μg/mL chlorpyrifos. Mosquitoes from Cosalapa and La Victoria were susceptible to organophosphates and to bendiocarb, but resistant to pyrethroids, with mortalities between 89% and 70% (WHO), and 88% and 78% (CDC), for deltamethrin and permethrin, respectively. High esterase levels are suggested as the resistance mechanism involved in the metabolism of pyrethroids in mosquitoes from both villages. Mosquitoes from La Victoria might also involve cytochrome P^450^. Therefore, organophosphates and carbamates are suggested to currently control *An. albimanus*. Its use might reduce the frequency of resistance genes to pyrethroids and vector abundance and may impede the transmission of malaria parasites.

## 1. Introduction

Malaria is a disease caused by parasites of the genus *Plasmodium*; *Plasmodium falciparum*, *P. vivax*, *P. malariae*, *P. ovale*, and *P. knowlesi*. These parasites are transmitted by mosquitoes of the genus *Anopheles*. The most fatal disease is caused by *P. falciparum* [1]. Prevention and control strategies focus on the fight against the mosquito vector, diagnostics, and the use of antimalarial drugs. Fortunately, malaria cases in Mexico have decreased considerably over the last 20 years (Figure 1) [2]. In 2021, malaria foci with recurrent transmission were reported only in 4 of the 32 states: Chiapas, Chihuahua, Sinaloa, and Campeche, and all infections were caused by *Plasmodium vivax* [3]. For 15 years, Mexico has been working on pre-elimination strategies, fulfilling the WHO goal of maintaining an incidence of fewer than one thousand cases per year. These strategies include the reinforcement of surveillance in first-level care units, the rapid analysis of thick blood smears, the monitoring of mosquito species, and the administration of treatments and fumigation. In addition, the population was recommended to install mosquito nets on doors and windows [4]. In 2021, Mexico reported 230 malaria autochthonous cases, of which 57% occurred in Chiapas. No deaths related to this condition have been reported since 1998 [2]. In addition to being a public health concern, malaria in Mexico comprises a social phenomenon of migration; in 2019, Mexico hosted more than half a million legal foreigners, a 58% increase since 2015 [5]. Chiapas is the southernmost state in the country, and the gateway to the migratory corridor, which represents a potential risk to increase malaria transmission or the introduction of new parasite strains [6]. Southern Mexico shares a border with Guatemala, which reported 1239 cases in 2021 [7]. El Salvador was fortunately certified malaria-free in 2021 [8]. Mexico also borders Belize, which was expected to obtain its certification by the end of 2022, but this has not yet been reported by WHO. Mexico plans to be certified as a malaria-free country by 2025; it will require new measures, strategies, and epidemiological surveillance to meet the goal.

Vector control is a strong arm of the surveillance system, and has been effective in preventing and reducing malaria transmission. The use of insecticide-impregnated bed nets and indoor residual spraying are the two main interventions [8]. Unfortunately, the control of malaria vectors has been threatened for years in the most affected areas of the world, due to the irrational use of insecticides. In 78 countries, anopheline species were found to be resistant to at least one class of insecticide used in public health, from 2010 to 2019, and in 29 of those countries, they were found to be multi-resistant to the four classes of insecticides (organochlorines, pyrethroids, organophosphates, and carbamates) [8]. Therefore, the management of resistance and knowledge of the resistance status to insecticide has become fundamental for successful vector control in malaria-endemic areas.

In 1997–1998, DDT was replaced by pyrethroids in Mexico, and these were used extensively for 15 years by vector control programs [9], until serious problems of vector resistance to insecticides were detected; after that, organophosphates and carbamates were used instead. At the Centro Regional de Investigación en Salud Pública (CRISP), we have records of resistance to insecticides from one of the main vectors of malaria [10] in the Soconusco region in southern Chiapas, a region which is important due to its agricultural and livestock activities. Although pyrethroids first saw use in public health in 1997, *Anopheles albimanus* collected in the banana-growing area from the Soconusco was already resistant to deltamethrin (75% mortality) and cyfluthrin (50% mortality) [11]. In 2003, resistance to deltamethrin was still present with mortalities of 86% and 75% in two villages of the banana growing area, Cosalapa (Suchiate, Municipality) and La Victoria (Mazatan, Municipality), respectively [12]. In Cosalapa, mosquitoes were also resistant to the organophosphate pirimiphos-methyl (78% mortality), but susceptible to the carbamate bendiocarb. Among the enzymes that metabolize insecticides, esterases and glutathione S-transferases (GSTs) were found to be slightly increased in mosquitoes from Cosalapa; and 24% of them had insensitive acetylcholinesterase (AChEi) to organophosphates and carbamates [13].

Currently, public health activities aimed at malaria vector control in this region are almost non-existent, since efforts are focused on following autochthonous malaria cases, which might not be the case for this region. The cases imported by migrants are left unattended because they are difficult to monitor due to their constant movement. However, local mosquitoes are usually affected by insecticides used in agriculture and livestock [11,14], rather than insecticides used for public health purposes. The purpose of this work was to update the susceptibility to insecticides and identify the resistance mechanisms present in the main malaria vector *An. albimanus* from villages of the Soconusco region in southern Chiapas, a region that comprises large movements of people migrating to enter the country. Thus, the optimal insecticides to reduce these malaria vector populations can be used by control programs.

## 2. Materials and Methods

### 2.1. Collection of Mosquitoes from the Soconusco Region

Two villages of the Soconusco, in southern Chiapas, Mexico, located in agricultural and livestock areas were visited during July and August 2022, and mosquitos were collected (Figure 2): Cosalapa (latitude 14°47′19.72″ N, longitude 92°17′12.44″ W) municipality of Suchiate, with eleven visits and 4078 caught mosquitoes, and La Victoria (latitude 14°48′19.59″ N, longitude 92°24′25.61″ W) municipality of Mazatán, with six visits and 5451 caught mosquitoes. Resting blood-fed *An. albimanus* were collected using hand-held mosquito aspirators. They were obtained from cattle corrals in Cosalapa, and sheep corrals in La Victoria. Female mosquitoes were placed in cages, transported to the insectary facility at CRISP, and supplied with cotton pads soaked in a 10% sugar solution. The determination of the species was corroborated following the keys of morphological characteristics according to Wilkerson [15].

To standardize age and physiological state, and prevent contamination by insecticides present in the area, we obtained the F_1_ generation of the field mosquito colonies. The insectary conditions were 27 ± 2 °C, 70–80% humidity, and 12:12 light-dark photoperiod.

### 2.2. Forced Oviposition of Blood-Fed Mosquitoes

Four days after blood-feed females were caught, forced oviposition was induced as reported previously by Dzul et al., 2007 [16] to obtain as many eggs as possible within 24 h.

Briefly, 5 to 10 gravid female mosquitoes were placed in 12-oz disposable plastic cups, covered with a cloth held in place with a rubber band. A plastic cup containing mosquitoes was placed in the freezer at −20 °C to sedate the insects. After five seconds, the plastic cup was gently shaken so that all the mosquitoes went down. After a second round of five seconds in the freezer, the sedated mosquitoes were placed onto a sheet of white paper. One or both pairs of mosquito wings were carefully amputated using a cutter. The recovered and standing mosquitoes were placed in 30 × 40 × 15 cm trays with water, whose internal walls were previously covered with filter paper (Figure 3), where they laid eggs for 24 h. The following day, mosquitoes were removed from the tray using forceps, the water was passed through filter paper to concentrate the mosquito eggs from the same village and date of collection, and the eggs were placed in clean trays with water. When the first instar larvae were observed, pools of 500 were redistributed into other trays containing 1 L of clean water to obtain F_1_ adults. On average, 15 trays were obtained per 1000 field-caught females. Larvae were fed with a Harlan 5001 protein diet passed through a 100-mesh sieve. The development from larvae to adults took 16–20 days. Two- to three-day-old female mosquitoes were used for the insecticide susceptibility bioassays, to calculate the mortality percentage for each insecticide. The other batches of mosquitoes were frozen to determine the enzyme levels, which was achieved using biochemical assays.

### 2.3. Insecticide Susceptibility Bioassays

The insecticides (technical grade, Sigma^®^, St. Louis, MO, USA) tested in the susceptibility tests were pyrethroids deltamethrin and permethrin, the organophosphates malathion and chlorpirifos, and the carbamate bendiocarb.

#### 2.3.1. WHO Tubes Tests

Female mosquitoes (F_1_) were subjected to bioassays following the methodology suggested by WHO, 2016 [17], using the diagnostic concentrations for the insecticides deltamethrin (0.05%), permethrin (0.75%), malathion (5%), and bendiocarb (0.1%). This method classifies the mosquito populations with percentages of mortality < 90 as resistant to the insecticide tested. Batches of 15–25 mosquitoes spent 1 h in four WHO testing tubes lined with Whatman’s No. 1 filter paper, which our laboratory previously impregnated with the diagnostic concentration of each insecticide. Another tube containing filter paper impregnated with olive oil, prepared with mosquitoes from the same batch, and exposed at the same time was used as control per set of insecticide tested. After exposure, mosquitoes were transferred to a holding tube containing 10% sucrose solution for 24 h under insectary conditions before mortality was scored.

#### 2.3.2. CDC Bottle Tests

To compare the results obtained with the WHO tube tests, mortality percentages were also calculated with the same field mosquito colonies and used to test the same group of insecticides following the methodology of the Center for Disease Control and Prevention [18].

##### Determination of the Diagnostic Concentration of the Insecticides

To test the susceptibility of the field mosquitoes, diagnostic concentrations for the four insecticides were calculated using mosquitoes from the susceptible reference strain. Batches of 15–25 mosquitoes from the susceptible Panama strain were exposed for 1 h, in four replicates of 250 mL Wheaton bottles, to five different concentrations of each insecticide that killed from 0 to 100% of the mosquitoes tested (for concentrations, see Table 1). Bottles were previously impregnated in our laboratory with different concentrations of insecticide diluted in 1 mL of acetone, as described in the CDC manual [18]. Other bottles impregnated with 1 mL acetone were prepared with mosquitoes from the same batch and exposed at the same time; one was used as a control per set of insecticides tested. After exposure, mosquitoes were transferred to a plastic cup, covered with netting, and provided with 10% sucrose solution for 24 h under insectary conditions before mortality was scored. Results of mortality percentages for each concentration of insecticide were submitted to the binary logistic regression model with QCal software [19], assuming the observed mortality curve adjusts to the model (*p* > 0.05); when rejected, the bioassay was repeated. The lethal concentrations that kill 99% of the mosquito population (LC_99_) were obtained for each insecticide. The diagnostic concentration for each insecticide was calculated by doubling the LC_99_ value (Table 1).

##### Mortality Percentages

The diagnostic concentrations calculated were used to coat a 250 mL Wheaton bottle dissolved in 1 mL of acetone. Then, 15–20 F_1_ female mosquitoes from the two field colonies (Cosalapa and La Victoria) were exposed per bottle in four replicates per set of insecticide. After 1 h of exposure, mosquitoes were transferred to plastic containers and maintained in the insectary to observe the mortality at 24 h.

The Panama strain of *An. albimanus* comprises mosquitoes susceptible to insecticides and were used as a control in both methodologies (WHO and CDC) every time a set with the diagnostic concentration was run. This colony strain has been maintained for more than 25 years without exposure to insecticides in the insectary at CRISP. Test results were adjusted for control mortalities with Abbott’s formula when required [20].

To determine the susceptibility/resistance levels of the field-caught mosquitoes, the percentage of mortality was calculated per insecticide and per village, and results were compared between both methods.

### 2.4. Biochemical Assays

The enzymes responsible for carrying out the detoxification of xenobiotics in organisms are synthesized by large families of genes [21]. Among the enzymes that metabolize insecticides are esterases, GSTs, and cytochromes P^450^, which could act together to metabolize large insecticide molecules. One of the most common mechanisms used to achieve this is overproduction; thus, they can metabolize or trap significant amounts of insecticide molecules to prevent them from reaching their site of action. To determine if the resistance of mosquitoes to insecticides is explained by the high levels of enzymes metabolizing insecticides, frozen F_1_ female mosquitoes from Cosalapa and La Victoria were processed by biochemical assays according to Penilla [14]. Each biochemical assay was repeated twice, with sets of 47 mosquitoes per village and per assay. A total of 94 mosquitoes per village were analyzed. If the results of a test were unreliable, they were excluded from the analysis.

Briefly, using a 96-well microplate, individual mosquitoes were homogenized in 200 µL of distilled water per well using a plastic pestle. Homogenates were centrifuged at 4000 rpm/4 °C/30 min and supernatants were dispersed in duplicates in clean microplates placed on ice, for α- and β-esterases (20 µL), ρ-nitrophenyl acetate (ρNPA)-esterases (10 µL), GST (10 µL), cytochromes P^450^ (20 µL), and to measure protein concentration (10 µL). Each microplate had two control wells with water instead of homogenates. Absorbance values of the chemical reactions were measured using a microplate reader Multiskan^®^ spectrum.

#### 2.4.1. α- and β-Esterase Assays

First, 200 µL of the sodium α-naphthyl acetate solution (100 µL of 30 mM α-NA in acetone added to 10 mL of 0.02 M phosphate buffer, pH 7.2) was deposited to the first duplicate of homogenate (α-esterases), and 200 µL of the sodium β-naphthyl acetate solution was deposited to the second duplicate (β-esterase). To stop the reaction, 50 µL of the fast blue dye solution was added (22.5 mg of fast blue in 2.25 mL of distilled water, 5.25 mL of 5% sodium lauryl sulfate diluted in 0.1 M sodium phosphate buffer pH 7.0) and the mixture was incubated at room temperature for 30 min. The reaction was measured at a wavelength of 570 nm at a fixed point. The results are reported as nmol of product generated per min/mg of protein.

#### 2.4.2. ρNPA-Esterases Assay

First, 200 µL of the substrate ρNPA (ρ-nitrophenyl acetate 100 mM in acetonitrile, and 50 mM phosphate buffer, pH 7.4, 1:100) was added to each well containing the mosquito homogenates. The ρNPA activity per individual was measured kinetically at a wavelength of 405 nm for 2 min, and the results are reported as µmol of the product/min/mg of protein using an extinction coefficient of 6.53 mM^−1^ (corrected for a path length of 0.6 cm).

#### 2.4.3. Glutathione-S-Transferases Assay

First, 200 µL of GSH/CDNB (10 mM reduced glutathione prepared in 0.1 M phosphate buffer, pH 6.5, and 63 mM chlorodinitrobenzene diluted in methanol) was added to each well containing the mosquito homogenates. The enzyme kinetics were measured at 340 nm for 5 min, and the GST activity is reported in mmol of conjugated CDNB/min/mg of protein, corrected for the path length and using an extinction coefficient of 9.6 mM^−1^.

#### 2.4.4. Cytochrome P^450^ Assay

First, 80 µL of potassium phosphate buffer (0.0625 M, pH 7.2) was added to each duplicate containing 20 µL of homogenate. This was followed by 200 µL of TMB solution (0.01 g of 3,3,5,5-tetramethylbenzidine diluted in 5 mL of methanol and mixed with 15 µL of 0.25 M sodium acetate buffer, pH 5.0), plus 25 µL of 5% hydrogen peroxide. The reaction was incubated for two hours at room temperature and measured at a wavelength of 650 nm at a fixed point. The optical density of each individual mosquito was compared to the standard curve of known concentrations of cytochrome P^450^ 2B4 from the rabbit. The results were reported in pmol of cytochromes P^450^/mg of protein.

#### 2.4.5. Protein Assay

An aliquot of 300 µL of the Bio-Rad solution (Dye Reagent Concentrated, BioRad, Hercules, CA, USA) [22] in a 1:4 dilution with distilled water, was added to each 10 µL duplicate of the homogenate. After 5 min of incubation at room temperature, the reaction was measured at 570 nm. The protein concentration per mosquito was determined using the standard curve derived from the bovine serum albumin.

The mean enzyme activity of each field mosquito population (village) was compared to that of the Panama strain and each other using the ANOVA, Kruskal–Wallis, and Dunnett’s test, with a significance of 95%.

## 3. Results

### 3.1. Insecticide Susceptibility Assays

#### 3.1.1. WHO Tube Tests

Using the WHO method [17] and its published diagnostic concentrations, mosquitoes from Cosalapa and La Victoria were found to be resistant to both pyrethroids, with 75% (*n* = 139) and 70% (*n* = 133) mortality for deltamethrin, and 89% (*n* = 150) and 85% (*n* = 180) for permethrin, respectively (Table 2). Mosquitoes from both villages were 100% susceptible to organophosphate malathion and to the carbamate bendiocarb.

#### 3.1.2. CDC Bottle Tests

The different concentrations to obtain a baseline causing mosquito mortalities from 0 to 100%, and the LC_99_ for each of the four insecticides following the CDC bottle bioassays, and using susceptible mosquitoes from the Panama strain are shown in Table 1. Deltamethrin ranged from 0.05 to 0.2 μg/mL/bottle, permethrin from 0.05 to 1.6 μg/mL/bottle, malathion from 0.5 to 3 μg/mL/bottle, and chlorpyrifos from 0.05 to 0.8 μg/mL/bottle.

The diagnostic concentrations of the four insecticides were obtained, doubling the LC_99_ value (Table 1): 0.7 μg/mL deltamethrin, 12 μg/mL permethrin, 14.4 μg/mL malathion, and 2 μg/mL chlorpyrifos. These diagnostic concentrations were used to determine the insecticide susceptibility of field mosquitoes. Mortality results for field-caught mosquitoes from Cosalapa and La Victoria were calculated as 81% and 85% for deltamethrin, and 78% and 88% for permethrin, respectively (Table 2). Mosquitoes from both Cosalapa and La Victoria were 100% susceptible to the organophosphates malathion and chlorpyrifos. No mosquitoes were available to test susceptibility with the carbamate bendiocarb.

Mortality percentages were very similar to those obtained by the WHO method (Table 2).

### 3.2. Biochemical Assays

#### 3.2.1. α-Esterases and β-Esterases

The results for individual mosquitoes for the concentration/content for each of the enzymes per mg of protein were calculated and grouped per village to obtain the mean values ± standard deviation (Sd), as shown in Figure 4. The comparison of the enzyme levels obtained between the two populations of field mosquitoes and the susceptible Panama strain is shown in the form of histograms (Figure 5). ANOVA results showed that the Mean (±Sd) values of α-esterases were significantly higher in mosquitoes from La Victoria (0.000579 ± 0.000412 µmol of α-naphthol/min/mg prot) than those from Cosalapa (0.000194 ± 0.000117 µmol of α-naphthol/min/mg prot) and those of the susceptible Panama strain (0.000132 ± 0.000056 µmol of α-naphthol/min/mg prot) (*p* < 0.001) (Figure 4). The same pattern was observed for β-esterases, with Mean (±Sd) values significantly higher in mosquitoes from La Victoria (0.000372 ± 0.000377 µmol of β-naphthol/min/mg prot) than those from Cosalapa (0.000101 ± 0.000104 µmol of β-naphthol/min/mg prot) or those of the susceptible Panama strain (0.000073 ± 0.000026 µmol of β-naphthol/min/mg prot) (*p* < 0.001).

While, mosquitoes from Cosalapa had only higher mean (±Sd) values of α-esterases (0.000194 ± 0.000117 µmol of α-naphthol/min/mg prot) and β-esterases (0.000101 ± 0.000104 µmol of β-naphthol/min/mg prot) than those from the Panama susceptible strain 0.000132 ± 0.000056 µmol of α-naphthol/min/mg prot (*p* < 0.001) and 0.000073 ± 0.000026 µmol of β-naphthol/min/mg prot (*p* < 0.01), respectively.

#### 3.2.2. Cytochromes P^450^

Mean (±Sd) values of cytochromes P^450^ were significantly higher in mosquitoes from La Victoria (0.231 ± 0.01658 pmol/mg protein) than those from Cosalapa (0.0138 ± 0.00822 of (pmol/mg prot) and, than those of the susceptible Panama strain (0.0145 ± 0.00582 (pmol/mg protein) (*p* < 0.001).

All other enzymes were at levels equal to, or lower than those of the susceptible mosquitoes used as a reference (Figure 4).

## 4. Discussion

Due to their lower toxicity, pyrethroids are the most used insecticides in Public Health in Latin America, so it might be common to find *An. albimanus* resistant to this toxicological group. In the south of the Yucatan peninsula, Mexico, resistance to deltamethrin was detected, along with elevated GSTs, cytochromes P^450^, and esterases [16]. Similar pyrethroid resistance and resistance mechanisms had already been reported in Guatemala [23]; in the indigenous region of Madugandí, Panama [24], and in Piura, Peru [25]; and the knockdown resistance (kdr) mutation selected by pyrethroid pressure was detected in *An. albimanus* from Nicaragua (1014C), Costa Rica (1014F) and Mexico (1014C) [26].

In the coastal plain of Chiapas, we have been documenting resistance to various pyrethroids and also to DDT in this malaria vector [11,14,27,28]. The resistance mechanisms were based on the high activity of esterases, and/or cytochromes P^450^s, and/or GSTs levels [13,27]. It was demonstrated that GSTs were a DDT metabolizer in this vector species [27]. Twenty years later, this study reports that resistance to pyrethroids is still present in *An. albimanus* collected in two villages in the Soconusco region, with 70% and 85% mortality. These results were corroborated by two diagnostic methods using WHO tubes and CDC glass bottles. However, unlike the previous findings, only α- and β-esterases were elevated in mosquitoes from both villages, suggesting that mosquitoes are capable of metabolizing pyrethroids. Presumably, the GST mechanism is not present because the resistance to DDT is reverted, as this insecticide is no longer used in public health [9]. Additionally, mosquitoes from La Victoria had elevated cytochromes P^450^, suggesting that these enzymes are involved in pyrethroid detoxification too. This was also suggested by the correlation obtained by the elevated levels of cytochromes P^450^ with resistance to pyrethroids in *An. albimanus* collected during four years of the insecticide resistance management program applied in the coastal plain of Chiapas [28].

On the contrary, the results suggest that *An. albimanus* from this region are not resistant to organophosphates and carbamates, and support their use to control pyrethroid-resistant mosquitoes in Cosalapa and La Victoria. Moreover, the spraying of pyrethroids with the synergist piperonyl butoxide (PBO) from commercial aerosol cans was effective against dengue vectors resistant to pyrethroids [29]. Therefore, pyrethroid/PBO mixtures might be used to control mosquitoes in areas with a similar resistant pattern to La Victoria. The alternation with organophosphates or carbamates could extend the usefulness of the mixtures of any pyrethroid and the PBO.

The Soconusco is a socioeconomic region located on the Pacific coastal plain and the Sierra Madre of Chiapas, associated with agricultural and cattle activities, and high activity in the use of insecticides. Practically all over the country, resistance to pyrethroids is the most serious problem on ranches, since they are regularly used as ixodicides [30]. Insecticide resistance in mosquitoes due to the excessive use of insecticides for the control of agricultural pests also is well documented [31]. Pyrethrin, for example, sees regulated use for the control of Banana rust thrips; this is carried out by the Federal Commission for the Protection against Sanitary Risks in Mexico [32]. Malaria control programs in Chiapas currently focus on seeking autochthonous and imported cases of malaria. At present, it is more likely that *An. albimanus* populations are being controlled and/or selected by the use of insecticides for agriculture and livestock, but not for public health purposes.

Since the 2000s, endemic transmission in Mexico is caused by *P. vivax*, which produces mild symptoms and rarely results in alarming symptoms. It is more difficult to eliminate due to its biological features, e.g., the presence of hypnozoites that cause relapse episodes [33]. The last death due to *P. falciparum* in the country occurred in 1998; however, the risk of re-emergency persists because of population migration from regions in which malaria transmission takes place, and more recently has significantly increased [6,34]. According to the Secretary of Health of Chiapas, more than 75,000 thick smear samples were taken in 2022 for the timely detection of malaria, and only 84 positive cases have been confirmed, which is the lowest throughout the entire history of this disease [35]. Other activities that claim to contribute to malaria reduction are the elimination and modification of vector habitats and breeding sites and the provision of bed nets with long-lasting insecticides. Chemical control also involves the use of larvicides in around 1500 locations with active transmission in the Lacandon forest and the municipalities of Tapachula and Tuxtla Gutiérrez mainly due to cases diagnosed in migrants. However, in the Soconusco region public health activities are minimal because there are few autochthonous cases of malaria; there is a constant risk of malaria introduction due to the frequent passage of migrants. Therefore, both control programs and farmers should make efforts to make proper use of chemicals in pest and vector control. Similarly, it is necessary to find and carry out effective epidemiological surveillance strategies with the migrant populations to keep the reemergence of malaria under control.

## 5. Conclusions

The resistance to pyrethroids was detected in *An. albimanus* in two villages from the Soconusco region with high agricultural and livestock activity, and a history of malaria transmission because of the high vector abundance. Alpha- and β-esterases were the predominant metabolism-based resistance mechanism, indicating the need to choose the best insecticide option for vector control and eliminate resistance genes that have endured for decades; this should be carried out by farmers/ranchers and by public health officials. This region has a constant risk for malaria introduction due to the frequent passage of migrants. Organophosphate and carbamate insecticides can be used for vector control with appropriate strategies and thus prevent the proliferation of the causative agent of malaria in the region.

## Figures and Tables

**Figure 1 ijerph-20-04258-f001:**
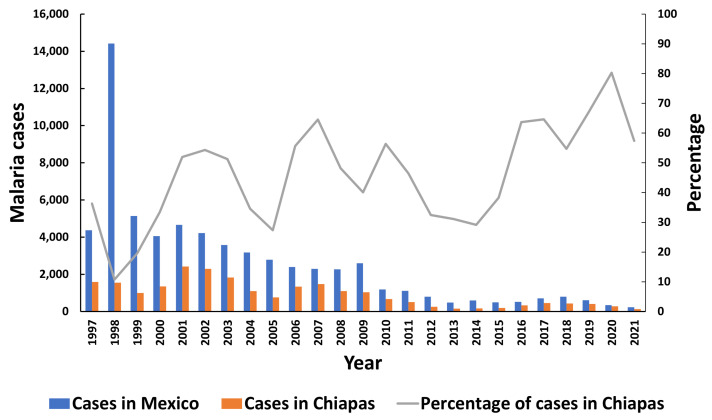
Number of malaria cases reported per year from 1997 to 2021 in Mexico (blue bars), and in Chiapas (red bars). The gray line represents the proportion of cases in Chiapas over the total number of cases in the country over a period of 24 years. Data obtained from the page of the Ministry of Health of the epidemiological bulletin https://www.gob.mx/salud/acciones-y-programas/historico-boletin-epidemiologico (accessed on 18 November 2022).

**Figure 2 ijerph-20-04258-f002:**
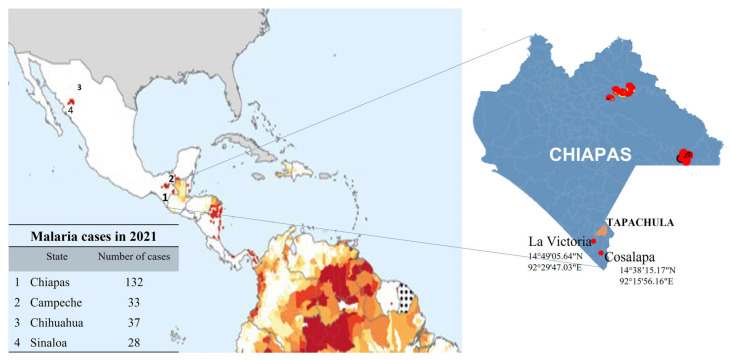
(**Left**): Map of Mexico indicating the four malaria foci (World Malaria Report 2022). (**Right**): two endemic malaria areas at the north and northeast of Chiapas are indicated; at the southern of Chiapas are shown Cosalapa and la Victoria villages from the Soconusco region, where the main malaria vector *Anopheles albimanus* was collected.

**Figure 3 ijerph-20-04258-f003:**
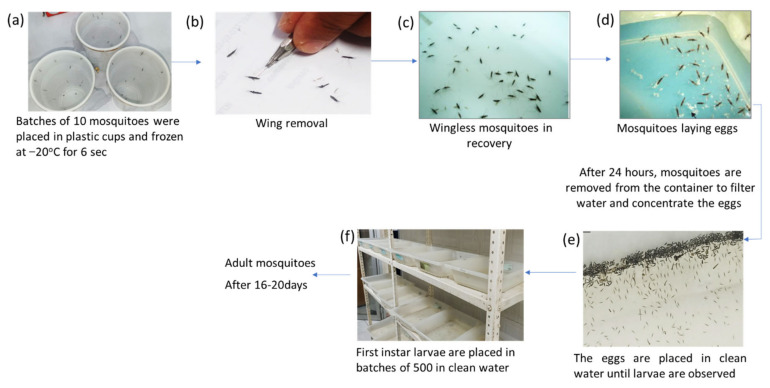
Procedure for forced oviposition of *Anopheles albimanus* under insectary conditions at the Centro de Investigación en Salud Pública, Instituto Nacional de Salud Pública, México. (**a**) Batches of 10 mosquitoes were placed in plastic cups and frozen at −20 °C for 6 s, (**b**) Wing removal, (**c**) Wingless mosquitoes in recovery, (**d**) Mosquitoes laying eggs, (**e**) The eggs are placed in clean water until larvae are observed, (**f**) First instar larvae are placed in batches of 500 in clean water.

**Figure 4 ijerph-20-04258-f004:**
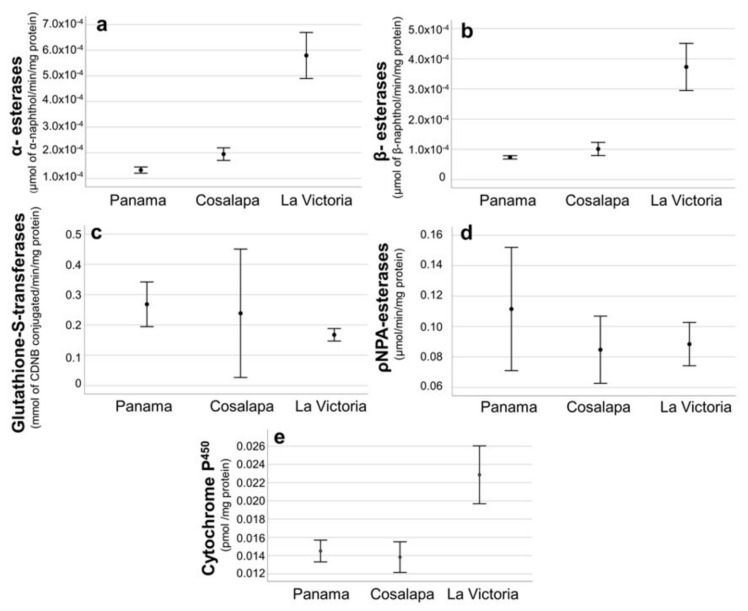
Means and standard deviations (SD) of the enzymes involved in the metabolism of insecticides in *Anopheles albimanus* collected in 2022 of the localities of Cosalapa and La Victoria in the Soconusco region of Chiapas, compared with the susceptible strain of Panama. (**a**): Mean (±Sd) of α-esterases, (**b**): Mean (±Sd) of β-esterases, (**c**): Mean (±Sd) of Glutathione S-transferases, (**d**): Mean (±Sd) of ρNPA-esterases, (**e**): Mean (±Sd) of cytochromes P^450^.

**Figure 5 ijerph-20-04258-f005:**
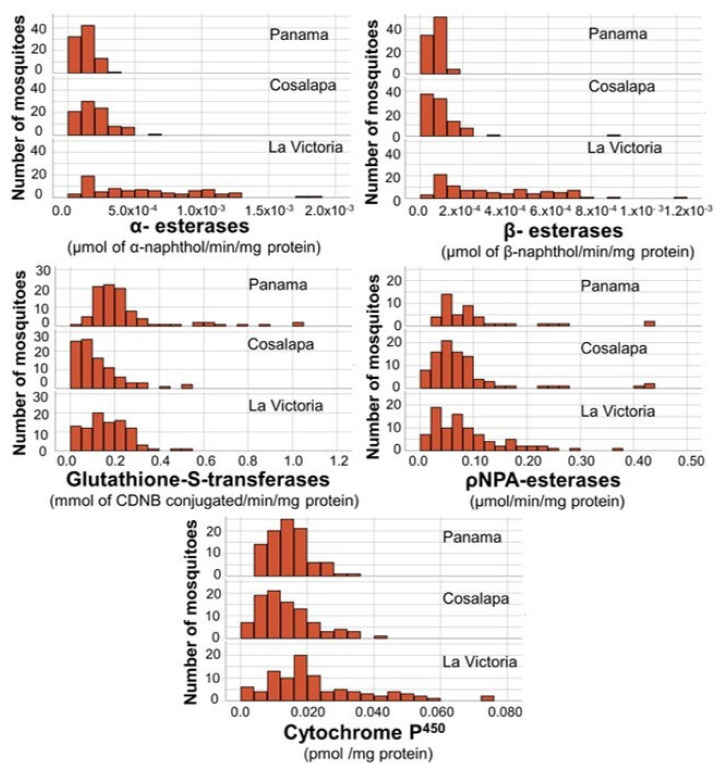
Histograms of the enzyme distribution between the two populations of field mosquitoes from Cosalapa and La Victoria, and the mosquitoes from the susceptible Panama strain. According to the ANOVA, the levels (Mean ± Sd) of α- and β-esterases, and cytochromes P^450^ were significantly higher in mosquitoes from La Victoria than mosquitoes from Cosalapa and those from the susceptible Panama strain (*p* < 0.001). While those from Cosalapa had higher α- and β-esterases than those from the Panama strain (*p* < 0.001).

**Table 1 ijerph-20-04258-t001:** Calculation of diagnostic concentrations for four insecticides based on double the lethal concentration at a 99% mortality of *Anopheles albimanus* from the Panama susceptible strain, according to the CDC bottle tests.

Insecticide	Concentrations in μg/mL/Bottle	Number of Mosquitoes	Lethal Concentration 99 (LC_99_)	Diagnostic Concentration (2LC_99_)
**Deltamethrin**	0.05, 0.075, 0.1, 0.15, 0.2	308	0.35	0.7
**Permethrin**	0.05, 0.1, 0.2, 0.4, 0.8, 1.2, 1.6	408	5.96	12
**Malathion**	0.5, 0.75, 1, 2, 3	253	7.2	14.4
**Chlorpyrifos**	0.05, 0.1, 0.2, 0.4, 0.8	360	0.99	2

The binary logistic regression model with QCal software [19] was used.

**Table 2 ijerph-20-04258-t002:** Percentages of mortality to pyrethoids and organophosphates of *Anopheles albimanus* collected in sheep and cattle corrals during July and August 2022 in Cosalapa and La Victoria villages in the Soconusco region, Chiapas.

WHO Susceptibility Test												
	Pyrethroids						Organophosphate			Carbamate		
	Deltamethrin			Permethrin			Malathion			Bendiocarb		
	0.05%			0.75%			5%			0.10%		
**Villages**	** *n* **	**dead**	**Mortality %**	** *n* **	**dead**	**Mortality %**	** *n* **	**dead**	**Mortality %**	** *n* **	**dead**	**Mortality %**
**Reference ***	201	201	100	199	199	100	210	210	100	201	201	100
**Cosalapa**	185	139	75	150	134	89	201	201	100	185	185	100
**La Victoria**	190	133	70	180	153	85	188	188	100	195	195	100
**CDC susceptibility test**												
	**Pyrethroids**						**Organophosphates**					
	**Deltamethrin**			**Permethrin**			**Malathion**			**Chlorpyrifos**		
	**0.7 µg/Bottle**			**12 µg/Bottle**			**14.4 µg/Bottle**			**2 µg/Bottle**		
**Villages**	** *n* **	**dead**	**Mortality %**	** *n* **	**dead**	**Mortality %**	** *n* **	**dead**	**Mortality %**	** *n* **	**dead**	**Mortality %**
**Reference ***	201	201	100	205	205	100	206	206	100	189	189	100
**Cosalapa**	101	82	81	115	90	78	112	112	100	100	100	100
**La Victoria**	122	104	85	103	91	88	101	101	100	101	101	100

* Susceptible colony from the Panama strain. Insecticide susceptibility tests were performed following the recommendations of the WHO and CDC, and were run together with mosquitoes susceptible to insecticides of the Panama strain.

## Data Availability

No data available.

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
