# Peer review of "Resistance to Pyrethroids in the Malaria Vector Anopheles albimanus in Two Important Villages in the Soconusco Region of Chiapas, Mexico, 2022"

_ijerph, 2023, doi:10.3390/ijerph20054258_

Round 1
Reviewer 1 Report
Here, the authors analyzed the insecticide susceptibility of An. albimanus collected in the Mexican state of Chiapas. This is an important study as vector control is a key component of malaria control in endemic areas. I had a difficult time going through the text. I strongly suggest that the authors work with a native speaker to prepare a thorough review of the manuscript. Apart from English issues, some sections must be better detailed before this manuscript is ready to be scientifically evaluated by peer review.
Major issues:
In its present form, it is unsuitable for publication and the reviewer comprehension and analysis are made difficult.
Material and Methods could be divided into subsections to make it easier to find relevant methodological information.
Provide more information on the protocols for insecticide resistance and enzymatic assays.
The description of the results should be better organized and more detailed. At the present form, the authors provide data in the form of tables and figures but the text lacks a more detailed description of these results.
The authors start by mentioning Table 2. Organize the text in order for the first mention to be of Table 1.
Font size should be increased. This is critical in Fig 2, 3 and 4
In Fig 4, what is the X-Axis?
How do Figure 4 and Table 3 complement each other?
Author Response
Major issues:
In its present form, it is unsuitable for publication and the reviewer comprehension and analysis are made difficult.
Response. We have tried to improve the manuscript, however, we have not had the time to send it for review by a native speaker, we hope it can be better understood. A new version has been prepared.
Material and Methods could be divided into subsections to make it easier to find relevant methodological information.
Response. Material and Methods have been divided into subsections.
Provide more information on the protocols for insecticide resistance and enzymatic assays.
Response. More infromation on the protocols have been provided, mainly for the enzyme assays.
The description of the results should be better organized and more detailed. At the present form, the authors provide data in the form of tables and figures but the text lacks a more detailed description of these results.
Response. The results were divided into sections, and the enzyme results with statistical differences were broken down in the Results section.
The authors start by mentioning Table 2. Organize the text in order for the first mention to be of Table 1.
Response. Table 1 is mentioned first but in the Material and Methods section.
Font size should be increased. This is critical in Fig 2, 3 and 4
Response. Fonts size from all Figs were increased.
In Fig 4, what is the X-Axis?
Response. The titles of the axis were added properly.
Reviewer 2 Report
The manuscript "Resistance to pyrethroids in the malaria vector Anopheles albimanus in two important villages in the Soconusco region of Chiapas, Mexico, 2022", needs a deep review to be suited for publication.
Here are a few considerations. Detailed comments are in the file provided.
Extensive editing of the English language and style are required to correct grammatical and concordance errors.
Lines 84-86: "(...) it is expected that these local mosquito are under selective pressure by insecticides used in agriculture and livestock than from insecticides used for public health purposes."
A similar statement is at the end of the discussion. However, the authors do not discuss what are the insecticides used for agriculture/livestock purposes in the areas evaluated that could be related to resistance to insecticides used for vector control strategies.
Lines 99-101: How the mosquitoes were collected?
How the mosquitoes collected were identified in order to assure that only An. albimanus were evaluated in the experiments?
Is this the sole species of Anopheles that occurs in the areas evaluated?
Lines 102-104: How many mosquitoes were collected per village?
Lines 140-141: "The levels of alpha-, beta-, and ρNPA- esterases, cytochromes P450, and GSTs were calculated."
Why these enzymes were chosen? What is their role in the mosquito system? The authors could provide a brief explanation of these aspects.
- An objective explanation of the methodology for each experiment performed must be provided. The authors only cite the sources but do not detail how and in what circumstances they performed their own experiments.
- A "Statistical Analysis" section is necessary to clarify how the data obtained were analyzed and how the statistical significance was achieved.
- Why a synergist-insecticide bioassay was not performed?

Author Response
The manuscript "Resistance to pyrethroids in the malaria vector Anopheles albimanus in two important villages in the Soconusco region of Chiapas, Mexico, 2022", needs a deep review to be suited for publication.
Here are a few considerations. Detailed comments are in the file provided.
Answers on the manuscript (pdf) are given in the same comment.
Extensive editing of the English language and style are required to correct grammatical and concordance errors.
Response. We have tried to improve the manuscript, however we have not had the time to send it for review by a native speaker, we hope it can be better understood.
Lines 84-86: "(...) it is expected that these local mosquito are under selective pressure by insecticides used in agriculture and livestock than from insecticides used for public health purposes."
A similar statement is at the end of the discussion. However, the authors do not discuss what are the insecticides used for agriculture/livestock purposes in the areas evaluated that could be related to resistance to insecticides used for vector control strategies.
Response. The sentences were reestructured through the manuscript. Two references were added in the Discussion section about the insecticides used in agriculture and in veterinary.
Lines 99-101: How the mosquitoes were collected?
Response. A sentence was added referring to the use of hand-held mosquito aspirators
How the mosquitoes collected were identified in order to assure that only An. albimanus were evaluated in the experiments?
Response. Mosquito species were checked using the key morphological characteristics according to Wilkerson, 1990. It was added to the new version of the manuscript.
Is this the sole species of Anopheles that occurs in the areas evaluated?
Response. Anopheles albimanus is very abundant in the villages collected, and is the primary vector in the coastal areas. The reference “Villarreal-Treviño et al 2020” where the distribition of the malaria vector in Mexico and the Soconusco area is disscused, was added to the new version of the manuscript.
Lines 102-104: How many mosquitoes were collected per village?
Response. The number of blood feed cought mosquitoes per village were added in the first paragraph subsection 2.1 of the new version of the manuscript.
Lines 140-141: "The levels of alpha-, beta-, and ρNPA- esterases, cytochromes P450, and GSTs were calculated."
Response. The description for each enzyme assay was detailed in the new version of the manuscript.
Why these enzymes were chosen? What is their role in the mosquito system? The authors could provide a brief explanation of these aspects.
Response. A brief explanation is provided at the beginning of the section 2.4 in the new version of the manuscript.
- An objective explanation of the methodology for each experiment performed must be provided. The authors only cite the sources but do not detail how and in what circumstances they performed their own experiments.
Response. The details of the procedures are explained for each enzyme in the new version of the manuscript.
- A "Statistical Analysis" section is necessary to clarify how the data obtained were analyzed and how the statistical significance was achieved.
Response. The statistical and sofwares used for the calculation of the LC99 and for the enzyme comparison were added in the new version of the manuscript.
- Why a synergist-insecticide bioassay was not performed?
Response. We prefer to do the biochemical assays instead, since the resistance mechanism can be detected at once, because it is very sensible to the detection of overproduced enzymes.
Please see the atachment of the manuscrpti in PDF, were you can find our answers

Round 2
Reviewer 1 Report
In this current version, the authors improved the text of the manuscript. In this present form, I’m afraid the text could still benefit from further proofreading by a native speaker. However, at this point, I believe this is an editorial decision. In its current form, it can mostly be understood by the reader. Apart from that, the manuscript science is sound and the description of the methodology has been improved.
- Ln 23-24 and others: “High esterase levels were the resistance mechanism involved in the metabolism of pyrethroids in mosquitoes from both villages “; “Mosquitoes from La Victoria also involved cytochrome P450“
Has this been confirmed? While these increased expressions for these enzymes are likely to result in the resistant phenotypes observed, the measurement of enzymatic activities does not provide final evidence, but a correlation. I suggest the authors tone down their current claims on the mechanism of resistance.
- Ln 26: “Its use will reduce the resistance gene frequency to pyrethroids “
Why is this the case?
- Table 3 could be converted into a figure.
Author Response
"Please see the attachment."

Reviewer 2 Report
The new version of the manuscript "Resistance to pyrethroids in the malaria vector Anopheles albimanus in two important villages in the Soconusco region of Chiapas, Mexico, 2022" has improved.
The authors addressed the points raised in the first version properly, and the data included in this new version has significantly increased the quality of the manuscript.
However, a language/writing review still needs to be performed, and other minor aspects still need attention. Please see the attached file.

Author Response
"Please see the attachment."
